# Evolving Resection Strategies for Non-Small Cell Lung Cancers: Translating Trial Evidence to Real-World Practice

**DOI:** 10.3390/cancers17213437

**Published:** 2025-10-27

**Authors:** Akshay J. Patel, Savvas Lampridis, Andrea Bille

**Affiliations:** 1Division of Thoracic Surgery and Lung Transplantation, Toronto General Hospital, 200 Elizabeth Street, Toronto, ON M5G 2C4, Canada; 2Institute of Immunology and Immunotherapy, University of Birmingham, Edgbaston, Birmingham B15 2TT, UK; 3Department of Thoracic Surgery, 424 General Military Hospital, 56429 Thessaloniki, Greece; 4National Heart and Lung Institute, Faculty of Medicine, Imperial College London, London SW7 2AZ, UK; 5Department of Thoracic Surgery, Guy’s Hospital, Guy’s and St. Thomas’ NHS Foundation Trust, Great Maze Pond, London Bridge, London SE1 9RT, UK; 6School of Cancer & Pharmaceutical Sciences, King’s College London, London WC2R 2LS, UK

**Keywords:** non-small cell lung cancer (NSCLC), sublobar resection, segmentectomy, lobectomy, robotic thoracic surgery, pulmonary metastasectomy

## Abstract

**Simple Summary:**

Surgery for early-stage lung cancer has traditionally involved removing an entire lobe of the lung. However, recent clinical trials suggest that removing a smaller portion of the lung, known as a segment or wedge, may be just as effective for very small tumours. This article discusses how these major studies have influenced current surgical thinking and examines why their results may not always reflect what happens in everyday clinical practice. It also explores how modern surgical technology, such as robotic systems, is changing the way surgeons perform these operations, and whether similar approaches could benefit patients with lung metastases from other cancers. The aim is to highlight where further research is needed to ensure that less invasive surgery continues to provide safe and effective cancer treatment.

**Abstract:**

Background: Lobectomy has long been the gold standard for early-stage NSCLC, but recent trials challenge its universality. The Japanese JCOG0802 trial demonstrated superior overall survival with segmentectomy versus lobectomy, whereas the North American CALGB140503 trial showed non-inferiority of sublobar resection, including wedge and segmentectomy, compared with lobectomy. Methods: This commentary critically evaluates evidence from JCOG0802 and CALGB140503 in the context of wider thoracic surgical practice. We examine trial disparities, the role of real-world data, heterogeneity in surgical approach and lymph node staging, the impact of robotics on segmentectomy adoption, and the application of segmental resection in pulmonary metastasectomy. Results: The divergent trial findings reflect differences in populations, nodal staging, and surgical definitions. Worldwide, variability in sublobar practice and inconsistent nodal assessment present challenges to oncological reliability. Robotics has facilitated a rapid increase in anatomical segmentectomy but risks shifting surgical intent from necessity to feasibility. In metastasectomy, segmentectomy may improve local control but remains unproven in randomised studies. Emerging strategies such as IVLP and molecular profiling offer potential to refine patient selection and outcomes. Conclusion: Sublobar resection represents a paradigm shift in the surgical management of small NSCLC. Ensuring oncological validity in real-world practice requires rigorous nodal staging, equitable access to technology, and prospective evaluation of segmentectomy in both primary and metastatic disease. Future advances will depend on aligning surgical precision with biologically informed patient selection.

## 1. Introduction

The surgical management of early-stage non-small cell lung cancer (NSCLC) has undergone a paradigm shift over the last decade. For much of the modern thoracic surgical era, lobectomy with systematic lymph node dissection was regarded as the gold standard for stage I disease, based on the landmark Lung Cancer Study Group trial of 1995, which reported higher rates of recurrence and cancer-related death following sublobar resection compared with lobectomy [1]. This dogma held sway for nearly three decades, despite significant advances in perioperative care, imaging, and minimally invasive surgery. The increasing detection of small peripheral lung nodules, driven by greater cross-sectional imaging and the advent of lung cancer screening programmes has re-opened the debate on whether lobectomy remains necessary for all early-stage tumours. Sublobar resection offers potential advantages such as the preservation of pulmonary function, reduced perioperative morbidity, and applicability in patients with marginal physiological reserve. However, the oncological adequacy of segmentectomy or wedge resection for tumours ≤ 2 cm has remained uncertain.

Two pivotal randomised controlled trials have provided key evidence in this regard. The Japanese JCOG0802/WJOG4607L trial demonstrated that segmentectomy achieved superior overall survival compared to lobectomy in patients with small peripheral NSCLC, albeit with increased local recurrence [2]. By contrast, the North American CALGB/Alliance 140503 trial reported that sublobar resection whether segmentectomy or wedge was non-inferior to lobectomy with respect to disease-free and overall survival [3]. These results have been widely hailed as practice-changing, yet they present nuanced and, at times, discordant messages regarding the relative merits of segmentectomy versus wedge resection, and their applicability to broader, more heterogeneous patient populations. The question remains, how representative of real-world patients are these trial cohorts?

The relevance of these trials to real-world practice is not straightforward. The Japanese trial enrolled a predominantly fit, non-smoking population with rigorous intraoperative nodal staging, while the CALGB trial incorporated both wedge and segmental resections and mandated frozen section analysis of lymph nodes prior to randomisation [2,3]. In contrast, global surgical practice demonstrates heterogeneity in surgical approach, inconsistent lymph node sampling, and marked variation in adoption of segmentectomy between institutions. The rapid uptake of robotic platforms has further complicated the landscape, facilitating technically complex segmentectomies that may be performed more for operative feasibility than for oncological necessity.

In parallel, there is renewed interest in anatomical segmentectomy for pulmonary metastasectomy, where deeper or anatomically challenging lesions may necessitate more radical parenchymal-sparing resections. Although segmentectomy offers the theoretical benefit of larger margins and nodal assessment, robust prospective data in the metastatic setting remain scarce. Emerging techniques, including isolated lung perfusion (IVLP), and upcoming datasets may further influence future practice in this domain.

This commentary will evaluate the evidence from JCOG0802 and CALGB140503, explore its translation to real-world thoracic surgical practice, and highlight areas where real-world data, technological advances, and pragmatic research are required to ensure oncological integrity is maintained in an era of evolving surgical strategy.

## 2. JCOG0802 vs. CALGB140503: A Tale of Two Trials

The JCOG0802 and CALGB 140503 trials [2,3] represent the most robust modern evidence comparing sublobar resection with lobectomy for small, peripheral NSCLC. Both have been hailed as practice-changing, yet their conclusions diverge, reflecting differences in design, population, and surgical philosophy.

JCOG0802/WJOG4607L [2], a multicentre Japanese trial, randomised 1106 patients with clinical stage IA NSCLC (tumours ≤ 2 cm, consolidation-to-tumour ratio > 0.5, located in the outer third of the lung) to lobectomy or anatomical segmentectomy. After a median follow-up of 7.3 years, segmentectomy yielded a statistically significant overall survival (OS) benefit compared with lobectomy (5-year OS: 94.3% vs. 91.1%; HR 0.663, *p* = 0.0082). Relapse-free survival (RFS) was equivalent, but local recurrence was nearly doubled after segmentectomy (10.5% vs. 5.4%). The authors attributed the OS advantage to better preservation of pulmonary function, allowing patients to tolerate subsequent treatments for recurrence or second primary tumours. Importantly, rigorous intraoperative lymph node assessment was mandated, and >95% of patients underwent systematic nodal dissection, factors that enhance oncological reliability but may limit generalisability to Western practice. CALGB140503 was an international trial involving 697 patients with peripheral, biopsy-proven NSCLC ≤ 2 cm and cN0 disease confirmed intraoperatively [3]. Patients were randomised to lobectomy or sublobar resection, which could be either wedge resection or anatomical segmentectomy, at the discretion of the surgeon. The trial demonstrated non-inferiority of sublobar resection with respect to both disease-free survival (5-year DFS: 63.6% vs. 64.1%; HR 1.01, 90% CI 0.83–1.24) and OS (5-year OS: 80.3% vs. 78.9%; HR 0.95, 95% CI 0.72–1.26). Locoregional recurrence rates were low and comparable between groups. The inclusion of wedge resections constituting around 60% of sublobar procedures has generated debate, as wedges are generally considered oncologically inferior to segmentectomy. Nonetheless, the non-inferiority conclusion suggests that for carefully staged peripheral tumours, wedge resection may suffice. A post hoc secondary analysis of the CALGB140503 randomised trial compared outcomes between lobectomy (LR), segmentectomy (SR), and wedge resection (WR) for peripheral cT1aN0 NSCLC. At 5 years, disease-free survival (DFS) was 64.7% for LR, 63.8% for SR, and 62.5% for WR (*p* = 0.888), while overall survival (OS) was 78.7%, 81.9%, and 79.7%, respectively (*p* = 0.873). Lung cancer–specific survival (LCSS) also showed no significant differences (86.8%, 89.2%, and 89.7%, respectively; *p* = 0.903). Locoregional recurrence occurred in 12% after SR and 14% after WR (*p* = 0.295). Six-month postoperative decline in FEV1 was minimal and comparable between SR (3%) and WR (5%, *p* = 0.930) [4].

The discordance between JCOG0802 and CALGB140503 is interesting. JCOG suggests segmentectomy is oncologically superior to lobectomy and CALGB indicates that lobectomy offers no significant survival advantage over sublobar resection including wedges. This could be explained by several factors. Firstly, population differences; JCOG enrolled predominantly fit, non-smoking Japanese patients, in whom second primaries may be more common and CALGB included a more heterogeneous Western cohort with higher smoking prevalence and comorbidity. The population difference is likely to result in a higher incidence of oncogene addicted cancers (EGFR positivity) which will result in a higher treatment rate with Tyrosine-kinase inhibitors. Secondly, the surgical technique and nodal assessment differed between cohort, in JCOG, >95% of patients had systematic lymphadenectomy, while in CALGB, nodal assessment varied despite mandated intraoperative sampling. In real-world Western settings, nodal sampling may be less rigorous, limiting the reliability of sublobar resections. The definition of sublobar resection also differed, JCOG excluded wedges, whereas CALGB included them, raising questions of whether segmentectomy alone should be considered the “true” comparator to lobectomy. Lastly, JCOG showed higher local recurrence after segmentectomy, but this did not translate into worse OS, possibly due to preserved lung function and improved salvage therapy tolerance. Supplementary multivariable analyses from the JCOG study [5] in the segmentectomy group revealed that pure-solid appearance on thin-section computed tomography (OR = 3.230; 95% confidence interval [CI]: 1.559–6.690; *p* = 0.0016), margin distance less than the tumour size (OR = 2.682; 95% CI: 1.350–5.331; *p* = 0.0049), and male sex (OR = 2.089; 95% CI: 1.047–4.169; *p* = 0.0366) were significantly associated with locoregional recurrence. Post hoc analyses [6] showed improved overall survival after segmentectomy in patients with pure-solid NSCLC compared with lobectomy. However, these outcomes were dependent on the patient’s age and sex with a need for further research to ascertain the clinically relevant indications for segmentectomy in radiologically pure-solid NSCLC. Large cohort data has shown that segmentectomy does not compromise overall survival regardless of the lobe from which the tumour was resected but it does associate with significantly less major morbidity compared to lobectomy [7]. Intersegmental tumours have been shown to display a higher rate of lymph node metastases yet no differences in 5-year overall and disease-free survival was seen between segmentectomy and lobectomy after propensity score-matching (1:1 *n* = 75 in each arm) [8]. However, the cohorts are small, and further information is required on induction treatment and patient and disease specific characteristics.

Taken together, the trials and allied cohort data challenge the entrenched dogma of lobectomy as the universal standard, but their applicability to the real-world setting is nuanced. The divergence highlights the importance of patient selection, lymph node staging, and surgical technique all of which are variably applied in modern day practice.

## 3. Lymph Node Staging in Sublobar Resection

Accurate lymph node staging remains central to the oncological validity of sublobar resection. The improved survival observed in JCOG0802 has been partly attributed to rigorous intra-operative lymphadenectomy, with systematic hilar and mediastinal sampling ensuring exclusion of occult N1 and N2 disease [2]. In contrast, CALGB140503 reported a notably higher rate of nodal upstaging after lobectomy compared with sublobar resection, raising concerns that inadequate nodal evaluation may compromise staging accuracy and long-term outcomes [3].

While guidelines recommend systematic nodal dissection or sampling during anatomical lung resections, adherence is variable. Audit data suggest that sublobar resections, particularly wedges, are frequently performed without adequate N1 or N2 assessment [9]. This practice risks misclassification of disease stage, inappropriate omission of adjuvant therapies, and underestimation of recurrence risk. The challenge is amplified by the technical nuances of segmentectomy. Dissection of intersegmental nodes requires meticulous identification and clearance of lymphatic stations, a task made more feasible with robotic technology [10]. Yet, the pressure of operative time, lack of training, and institutional culture can result in abbreviated or omitted nodal assessment. The UK National Lung Cancer Audit (NLCA) has repeatedly identified variation in nodal dissection practice, highlighting a quality gap that becomes particularly relevant as sublobar resections increase [9]. Emerging technologies may offer solutions. Techniques such as intraoperative frozen section of segmental nodes or molecular staging with one-step nucleic acid amplification (OSNA) can rapidly assess N1 involvement, guiding conversion to lobectomy where appropriate [11]. Similarly, advances in intraoperative navigation and fluorescence imaging may improve nodal detection. However, these innovations remain inconsistently adopted and are not yet integrated into all surgical guidelines.

Ultimately, the integrity of sublobar resection depends not only on the parenchymal margin but also on the thoroughness of nodal assessment. The JCOG trial’s meticulous approach is difficult to replicate in real-world practice, and unless systematic lymphadenectomy becomes embedded into surgical culture, there is a risk that segmentectomy outcomes will fall short of those reported in trials. Ensuring robust nodal staging should therefore be considered the Achilles’ heel and litmus test of sublobar resection programmes worldwide.

Allied to this, access to tissue is the key to ensuring the correct treatments are allocated to patients. In the current era, with molecular stratification of oncogene-addicted cancers and next generation sequencing to better understand disease biology, there is an increasing need to ensure robust nodal dissection and tissue sampling is performed. We are seeing a growing trend of small stage I cancers recurring within 12 months which highlights two things, firstly contemporary staging systems are grossly imperfect and secondly, tumour biology is the most critical factor to determining disease behaviour and unfortunately the parameter which is least understood. If segmental resection or indeed sublobar resection cannot adequately ensure adequate disease margins or procurement of enough nodal tissue, lobectomy has to remain the fallback option and the “gold standard” for tumour resection when the primary metric is oncological clearance.

## 4. The Robotic Revolution

The advent of robotic-assisted thoracic surgery (RATS) has significantly reshaped the landscape of lung cancer surgery in the UK. While the uptake of video-assisted thoracoscopic surgery (VATS) was gradual and uneven across centres, the adoption of robotics has been rapid, particularly in high-volume institutions. This expansion has coincided with increasing interest in sublobar resections, particularly anatomical segmentectomy. Robotic systems provide several advantages over conventional VATS that are particularly relevant for anatomical segmentectomy. Three-dimensional visualisation, enhanced instrument articulation, and stable camera control allow precise dissection of segmental vessels and bronchi [12]. Techniques such as indocyanine green (ICG) fluorescence imaging for delineation of the intersegmental plane are now routine in many robotic centres, reducing technical barriers that previously limited adoption of segmentectomy [13]. The net result has been a rapid increase in robotic segmentectomies in centres where platforms are available. However, this technological facilitation raises critical questions. There is a risk that the ease of performing complex segmentectomies robotically may encourage their use in situations where oncological necessity is debatable, for example, in small, peripheral lesions that could be adequately treated with wedge resection. Furthermore, while robotics enables meticulous anatomical resection, it has not yet been shown to improve oncological outcomes compared with VATS [14,15,16], and the survival benefit reported in JCOG0802 cannot be directly attributed to technology [2].

Another concern is the shift in surgical intent, in some centres, segmentectomy may be chosen less for oncological indication and more because robotics makes the operation straightforward and reproducible. This blurring of rationale risks undermining the principle that the extent of resection should be determined by tumour biology and staging, not solely by technical feasibility.

From a training and workforce perspective, the introduction of robotics also has implications. Robotic segmentectomy requires different skill sets and team structures compared with VATS, and its concentration in high-resource centres risks exacerbating geographic variation in access to sublobar resection. The Society for Cardiothoracic Surgery UK (SCTS) Robotic Training Survey (SORTS UK) is an initiative that has recently been developed, and the aim is to highlight any variability in training exposure and to support the need for structured pathways to ensure equitable access to robotic expertise. Robotics has undoubtedly catalysed the expansion of anatomical segmentectomy by lowering technical barriers and improving precision. Yet, without robust real-world outcome data and standardised indications, there is a danger that robotic technology may confound surgical decision-making, prioritising technical elegance over oncological necessity.

## 5. Segmentectomy in Pulmonary Metastasectomy: A New Paradigm?

Pulmonary metastasectomy (atypical resections) remains one of the most common thoracic oncological operations worldwide, with evidence suggesting a survival benefit in carefully selected patients [17]. This is however historical and retrospective data. There is no randomised data in this setting to show any meaningful survival benefit of metastasectomy and indeed this was further reinforced by the failure of the PulMICC trial [18,19]. Historically, the default approach has been wedge resection, balancing parenchymal preservation against oncological adequacy. Segmentectomy, however, has gained traction as a potential middle ground, offering anatomical clearance of segmental nodes and bronchovascular structures while avoiding the morbidity of lobectomy. Retrospective series and registry analyses suggest that segmentectomy may provide superior local control compared with wedge resection, particularly for centrally located metastases or those abutting segmental structures [20,21,22]. Reports demonstrate lower local recurrence rates when segmentectomy is performed, with similar OS to wedge in most series [20]. However, robust randomised data are lacking, and practice remains surgeon- and institution-dependent. The PulMiCC trial, the only randomised controlled study of pulmonary metastasectomy (in colorectal cancer), highlighted that the survival benefit of surgery may be more modest than previously assumed [18,19]. This finding underscores the need for better patient selection and careful evaluation of the true oncological advantage of segmentectomy over wedge.

Segmentectomy is technically more demanding, requiring precise identification of intersegmental planes, bronchovascular anatomy, and nodal drainage. Robotic technology has facilitated this through enhanced three-dimensional visualisation and improved instrument articulation, potentially expanding the use of segmentectomy for metastases that might otherwise have been managed with a lobectomy. Conversely, for small, peripheral, and multiple metastases, wedge remains quicker, less invasive, and parenchymal sparing, a crucial factor in patients with limited pulmonary reserve.

Novel strategies are being investigated to enhance outcomes in pulmonary metastasectomy. Isolated in vivo lung perfusion (IVLP), which allows high-dose regional chemotherapy delivery during surgery, has shown promise in phase I/II dose escalation trials whilst allowing from prolonged perfusion without acute lung injury [23,24]. Further data has demonstrated that the trend of recurrences is less in the IVLP treated lung, despite greater disease burden at baseline. With this platform, it is possible to safely deliver chemotherapy doses far exceeding safe systemic doses [25]. In parallel, advances in molecular profiling, circulating tumour DNA (ctDNA), and minimal residual disease (MRD) detection may help identify patients most likely to benefit from surgery versus systemic or locoregional alternatives.

## 6. Future Directions and Research Needs

Future research should focus on prospective, real-world studies that evaluate sublobar resections in more diverse populations and practice settings. Efforts to standardise intraoperative lymph node staging remain essential to preserve oncological integrity outside of the trial environment. The increasing use of robotic platforms warrants dedicated studies to determine whether their facilitation of anatomical segmentectomy translates into measurable oncological or functional benefit, rather than reflecting only technical feasibility. In pulmonary metastasectomy, well-designed trials are needed to clarify whether segmentectomy confers superior local control or survival compared with wedge resection, particularly when integrated with molecular profiling and minimal residual disease assessment.

## 7. Conclusions

Sublobar resection has now been established as an acceptable alternative to lobectomy for small, peripheral NSCLC, supported by high-quality randomised evidence. However, differences in outcomes between JCOG0802 and CALGB140503 highlight how patient selection, surgical technique, and lymph node staging profoundly influence results. Importantly, the strict protocols of these trials do not fully reflect the variability and complexity of real-world surgical practice. Extending the benefits of sublobar resection beyond the trial setting will require meticulous staging, adherence to oncological principles, and careful integration of new technologies. As evidence evolves, a balance between parenchymal preservation and oncological radicality will remain central to optimising patient outcomes.

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
