# Peer review of "Evolving Resection Strategies for Non-Small Cell Lung Cancers: Translating Trial Evidence to Real-World Practice"

_cancers, 2025, doi:10.3390/cancers17213437_

Round 1

Reviewer 1 Report

Comments and Suggestions for Authors

The manuscript is well written and deals with highly relevant and timely topics. The cited literature and the issues addressed are of great interest, particularly for surgeons approaching this area of work. However, the commentary would benefit from a clearer logical flow between sections, as well as completion of unfinished parts and the inclusion of additional supporting references.  At present, the manuscript progresses from sublobar resections to robotic surgery, then to lymph node dissection, and finally to metastasectomy. This order feels somewhat fragmented and disrupts the coherence of the commentary. A more logical structure would be to begin with sublobar resections (segmentectomy versus atypical resections), then move to nodal staging and the role of lymphadenectomy, and finally address robotics and future directions. This would create a smoother narrative and improve readability.  Some areas of the manuscript appear unfinished, with empty placeholders left from the template. These should be completed to provide a polished final version. In addition, expanding the level of detail in certain sections would help strengthen the analysis. The article would benefit from the inclusion of additional references to reinforce the authors’ arguments and to situate their opinions within the wider body of scientific evidence. Finally, ensure consistency in terminology when discussing sublobar resections (segmentectomy vs atypical resections) and double-check formatting, as the presence of draft elements gives the impression of an incomplete submission.

Author Response

RESPONSE: Thank you for the detailed review and evaluation of our paper. We have reordered the narrative as per your suggestion and we have expanded on the relevant sections as directed.

Reviewer 2 Report

Comments and Suggestions for Authors

The structure of the review is inadequate. Furthermore, the existing related papers analyzed lack sufficient content and are insufficient to draw appropriate conclusions. Surgical approaches vary depending on the hospital, surgeon, and surgical environment, and this description is too superficial.

Author Response

RESPONSE: Thank you for the detailed review and evaluation of our paper. We have reordered the narrative as per your suggestion and we have expanded on the relevant sections. However, the feedback was quite vague and not particularly insightful. The aim of the review was to provide a brief commentary on the status of SLR in the current era based on the best evidence available without needing to cite every retrospective cohort study published.

Reviewer 3 Report

Comments and Suggestions for Authors

This is a well-organized commentary on surgical strategies for peripheral NSCLC and metastatic lung tumors. For peripheral NSCLC sized 2 cm or less, sublobar resection has become the standard of care based on two pivotal studies. The commentary also addresses issues relevant to the real-world settings, such as wedge resection versus segmentectomy and systematic lymph node dissection, and the recent introduction of robotic-assisted thoracic surgery. I have no major additional comments; however, it might be even more valuable if the authors could also discuss whether treatment strategies for these early-stage lung cancers should take into account the biological background, including genetic alterations and PD-L1 expression.

Author Response

RESPONSE: Thank you for the detailed review and evaluation of our paper. We have included a section on early-stage lung cancers and have taken into account the biological background, including genetic alterations and PD-L1 expression.

Round 2

Reviewer 1 Report

Comments and Suggestions for Authors

The authors have made substantial structural revisions to the manuscript, which have significantly improved its overall quality. The revised version shows clearer organization, enhanced methodological coherence, and a more focused discussion of the biological and clinical implications. The modifications have considerably strengthened the scientific value and readability of the work

Author Response

Thank you for your kind comments and very useful revision suggestions.